# Complete Chloroplast Genome Sequence and Phylogenetic Analysis of *Quercus bawanglingensis* Huang, Li et Xing, a Vulnerable Oak Tree in China

**Xue Liu** [1], **Er-Mei Chang** [1], **Jian-Feng Liu** [1,*], **Yue-Ning Huang** [1], **Ya Wang** [1], **Ning Yao** [1] **and Ze-Ping Jiang** [1,2]

[1] Key Laboratory of Tree Breeding and Cultivation of State Forestry Administration, Research Institute of Forestry, Chinese Academy of Forestry, Beijing 100091, China

[2] Research Institute of Forest Ecology, Environment and Protection, Chinese Academy of Forestry, Beijing 100091, China

*   Correspondence: liujf2000cn@163.com

**Abstract:** *Quercus bawanglingensis* Huang, Li et Xing, an endemic evergreen oak of the genus *Quercus* (Fagaceae) in China, is currently listed in the Red List of Chinese Plants as a vulnerable (VU) plant. No chloroplast (cp) genome information is currently available for *Q. bawanglingensis*, which would be essential for the establishment of guidelines for its conservation and breeding. In the present study, the cp genome of *Q. bawanglingensis* was sequenced and assembled into double-stranded circular DNA with a length of 161,394 bp. Two inverted repeats (IRs) with a total of 51,730 bp were identified, and the rest of the sequence was separated into two single-copy regions, namely, a large single-copy (LSC) region (90,628 bp) and a small single-copy (SSC) region (19,036 bp). The genome of *Q. bawanglingensis* contains 134 genes (86 protein-coding genes, 40 tRNAs and eight rRNAs). More forward (29) than inverted long repeats (21) are distributed in the cp genome. A simple sequence repeat (SSR) analysis showed that the genome contains 82 SSR loci, involving 84.15% A/T mononucleotides. Sequence comparisons among the nine complete cp genomes, including the genomes of *Q. bawanglingensis*, *Q. tarokoensis* Hayata (NC036370), *Q. aliena var. acutiserrata* Maxim. ex Wenz. (KU240009), *Q. baronii* Skan (KT963087), *Q. aquifolioides* Rehd. et Wils. (KX911971), *Q. variabilis* Bl. (KU240009), *Fagus engleriana* Seem. (KX852398), *Lithocarpus balansae* (Drake) A. Camus (KP299291) and *Castanea mollissima* Bl. (HQ336406), demonstrated that the diversity of SC regions was higher than that of IR regions, which might facilitate identification of the relationships within this extremely complex family. A phylogenetic analysis showed that *Fagus engleriana* and *Trigonobalanus doichangensis* form the basis of the produced evolutionary tree. *Q. bawanglingensis* and *Q. tarokoensis*, which belong to the group *Ilex*, share the closest relationship. The analysis of the cp genome of *Q. bawanglingensis* provides crucial genetic information for further studies of this vulnerable species and the taxonomy, phylogenetics and evolution of *Quercus*.

**Keywords:** chloroplast (cp) genome; *Q. bawanglingensis*; comparative analysis; phylogenetics; interspecific relationships

---

## 1. Introduction

The cp genomes of most gymnosperms are uniparentally paternally inherited, whereas the majority of angiosperms are uniparentally maternally inherited [1]. In most angiosperms, the cp genomes, which encode approximately 130 genes and range from 76 to 217 kb [2,3], are typical double-stranded circular DNA composed of four regions containing two copies of inverted repeats (IRa and IRb) and two

single-copy regions (LSC and SSC) [4,5]. Due to its uniparental inheritance, highly conserved structure, general lack of recombination and small effective population size, the analysis of cp DNA has been deemed a useful method for evolution research and the exploration of plant systematics [6–9]. In fact, the availability of sufficient data on cp genomes is crucial for phylogenetic relationship reconstruction, i.e., the assessment of relationships within angiosperms [10–12], the identification of members of Pinaceae [13] and *Pinus* [14], and adequate comparisons, i.e., cp genomes from sister species [15] and possibly multiple individuals [16]. At present, approximately 3000 plastid genomes of Eukaryota are shareable in the National Center for Biotechnology Information database (NCBI; Available online: https://www.ncbi.nlm.nih.gov/genomes/GenomesGroup.cgi?opt=plastid&taxid=2759&sort=Genome) due to improvements in sequencing technologies. In addition, molecular genetic methodologies based on nuclear and organellar genomes are crucial for conservation studies [17], particularly the conservation of threatened species for which there is scarce information on the genetic variation among populations [18]. Comprehensive analysis of both cpDNA and nDNA sequences could provide supplementary and often contrasting information on the genetic diversity among populations [17,19–21], which could be used to explore the causes of species threats and for the formulation of appropriate conservation measures. In addition, the DNA barcode has broad applications for rapid and accurate species identification [22]. Although the design of universal primers for single-copy nuclear sequences related to species boundaries is difficult, these nuclear primers might also be used for species discrimination in the future [23]. Barcodes based on whole plastid analyses that show interspecific discrepancies are expected to yield more information at the species and population levels for species identification to reveal new species and aid in biodiversity surveys and thus offer useful conservation strategies [24,25].

Oaks (*Quercus* L., Fagaceae) encompass approximately 500 species that are located throughout the northern hemisphere, although they mainly thrive in northern South America and Indonesia [26,27] and are dominant, diverse angiosperm plants due to their economical, ecological, religious and cultural benefits [28]. In biology research, oaks are widely used for both hybridization and introgression. Oaks were originally formalized as belonging to the subgenera *Cyclobalanopsis* and *Quercus* based entirely on their morphological characteristics [27,29], and the shift from a morphology- to a molecular-based classification changed the classification of oaks to the following two major clades (each comprising three groups): a Palearctic-Indomalayan clade (group *Ilex*, group *Cerris* and group *Cyclobalanopsis*) and a predominantly Nearctic clade (group *Protobalanus*, group *Lobatae* and group *Quercus*) [28]. Most recently, based on their morphology, molecular features, and evolutionary history, *Quercus* was split into two subgenera, *Quercus* and *Cerris*, and these subgenera include eight groups: for subgenus *Quercus*, group *Protobalanus*, group *Ponticae*, group *Virentes*, group *Quercus* and group *Lobatae*, and for subgenus *Quercus*, group *Cyclobalanopsis*, group *Ilex*, and group *Cerris* [30]. In subsequent studies, the main challenge in oak classification will be infrasectional classification. With the rapid development of sequencing technology, genomic databases are becoming increasingly vital for in-depth studies of plant phylogenetics [31,32]. However, due to the use of plastid and nuclear data, incongruent phylogenies have been observed in not only *Quercus* but also other genera [33–35]. In fact, high-resolution phylogenomic approaches can be used to assess the nuclear genome (e.g., RAD-sequencing) and likely provide even more highly important sources of information for phylogenetic and evolutionary studies, particularly in American oaks, such as *Lobatae*, *Protobalanus* and *Quercus* [36–38]. Plastid genomes are also important, because they can provide supplementary information that can be somewhat hidden in nuclear genomes (e.g., population–area relationships, ancient taxa histories and relationships) [38–40]. Hence, it is necessary to obtain preliminary cp genome data that can be used in future studies for species identification, for the assessment of relationships and eventual phenomena, such as reticulation, isolation, and introgression, and for establishing adequate conservation strategies.

*Q. bawanglingensis* is an endemic and vulnerable plant in China that is included in the Red List of Chinese Plants at the D2 VU level (the Red List of Chinese Plants. Available from: http://www.chinaplantredlist.org) [41] based on the following criteria: a decline in the area of occupancy

(AOO) by <20 km$^2$ or in the number of locations by ≤5. Nevertheless, its genetic background and resources have not been widely studied. Deng et al. (2017) reported that *Q. bawanglingensis*, which belongs to the phylogenetic group *Ilex Q. setulosa* complex, was more related to *Q. setulosa* in terms of leaf epidermal features [42]. As recorded in the Flora of China (the Flora of China. Available from: http://foc.iplant.cn), *Q. bawanglingensis* is considered a distinct species related to *Q. phillyreoides*, but its genetic traits and taxonomic status are uncertain. Thus, a high-resolution and supported molecular phylogenetic tree is necessary. Obtaining cp genome information is necessary due to the lack of data on *Q. bawanglingensis*, and the importance and availability of information on the plastid genomes of oaks for detailed comparisons are increasing [40,43–46]. Polymorphic chloroplast microsatellite markers designed based on a cp genome analysis can be utilised to comprehend the levels and patterns of the geographical structure and genetic diversity of *Q. bawanglingensis*, and this information can subsequently be used formulate an effective protection strategy.

In this study, we first sequenced and described the complete cp genome of *Q. bawanglingensis* and performed a comparative analysis of the cp genomes of multiple *Quercus* species in order to (1) investigate the structural patterns of the whole chloroplast genome of *Quercus* species including the genome structure, gene order and gene content; (2) examine abundant simple sequence repeats (SSRs) and large repeat sequences in the whole cp genome of *Q. bawanglingensis* to provide markers for phylogenetic and genetic studies; and (3) construct a chloroplast phylogeny for Fagaceae species using their whole cp DNA sequences.

## 2. Materials and Methods

### 2.1. Chloroplast DNA Extraction, Illumina Sequencing, Assembly, Annotation and Sequence Analyses

A single individual of *Q. bawanglingensis* (height: 3.3 m, diameter at breast height (DBH): 7.8 cm) was used as a sampling object from Mount Exianling (109°06′, 35.88″E; 19°00′, 45.65″N) on Hainan Island (Figure A1) [47]. Hainan, a portion of the Indo-Burma Biodiversity Hotspot and the second largest island in China, is located at the northern edge of tropical Southeast Asia. Mount Exianling, the largest and the best-preserved tropical limestone rainforest on Hainan Island, is situated in the western area of this island [47]. The mount covers 2000 ha. and has an altitude from 100 to 1238 m. The island is characterised by a typical tropical monsoon and continental climate, with a rainy season (May to November) and a dry season (December to April of the following year). The annual average temperature is 24.5 °C, and the annual precipitation is 1647 mm.

Fresh leaves of the individual were collected and flash-frozen in liquid nitrogen and then stored in a refrigerator (−80 °C) until DNA extraction. DNA extraction was performed using the modified CTAB method [48]. DNA quality was assessed in a one drop spectrophotometer (OD-1000, Shanghai Cytoeasy Biotech Co., Ltd., Shanghai, China), and integrity was evaluated using 0.8% agarose gel. Sequencing was performed using an Illumina Hiseq4000 platform (Genepioneer Biotechnologies Co. Ltd., Nanjing, China) with PE250 based on Sequencing by Synthesis (SBS), with at least 5.74 GB of clean data obtained for *Q. bawanglingensis*. We then used FastQC v0.11.3 to trim the raw reads, and the cp-like reads were then extracted through a BLAST analysis between the trimmed reads and references (*Q. tarokoensis* and *Q. tungmaiensis*). We subsequently assembled the sequences with the cp-like reads using NOVOPlasty [49]. Genome annotation was performed using CPGAVAS [50], and the results were checked using DOGMA (DOGMA. Available from: http://dogma.ccbb.utexas.edu) and BLAST [51]. The tRNAs were identified by tRNAscan-SE [52], and we then mapped the entire genome using the OGDRAWv1.2 programme (OGDRAWv1.2. Available from: http://ogdraw.mpimp-golm.mpg.de) [53]. The cp genome sequences of *Q. bawanglingensis* have been deposited in GenBank (MK449426). SSRs and long repeats were determined using the MIcroSAtellite (MISA) identification tool (MISA. Available from: http://pgrc.ipk-gatersleben.de/misa/misa.html) [54] and REPuter (REPuter. Available from: https://bibiserv.Cebitec.uni-bielefeld.de/reputer) [55]. We also conducted various analyses of the

guanine and cytosine (GC) content, codon usage, diversification in synonymous codon usage, and relative synonymous codon usage (RSCU) values.

## 2.2. Genome Comparison

Paired sequence alignment was performed using MUMmer [56]. mVISTA [57] was used to examine the genetic divergence among nine complete cp genomes, namely, those of *Q. bawanglingensis*, *Q. tarokoensis* (NC036370.1), *Q. aliena var. acutiserrata* (KU240008), *Q. baronii* (KT963087), *Q. aquifolioides* (KX911971), *Q. variabilis* (KU240009), *Fagus engleriana* (KX852398), *Lithocarpus balansae* (KP299291) and *Castanea mollissima* (HQ336406), in the Shuffle-LAGAN mode [58] with the genome of *Q. tarokoensis* as the reference genome. The cp genome sequences of *Q. bawanglingensis*, *Q. tarokoensis*, *Q. aliena var. acutiserrata*, *Q. baronii*, *Q. aquifolioides* and *Q. variabilis* were aligned using MAFFT v.5 [56], and a sliding window analysis was performed to detect the nucleotide diversity of the cp genomes using DnaSP v5 [59].

## 2.3. Phylogenetic Analysis

The phylogenetic analysis was performed using FastTree based on sequences from 29 taxa, namely, 24 Fagaceae species, three Betulaceae species and two outgroups (*Populus trichocarpa* and *Theobroma cacao*), all of which were downloaded from the NCBI except those of *Q. bawanglingensis*. MAFFT v.5 [56] was utilized to align the cp genomes of the 29 species. We also performed multiple sequence alignments manually using BioEdit [60] and reconstructed a maximum likelihood (ML) tree using FastTree version 2.1.10 [61].

## 3. Results

### 3.1. Features of the Chloroplast Genome of Q. bawanglingensis

In total, at least 5.74 GB of clean data was obtained for *Q. bawanglingensis*, and these data were assembled into a double-stranded circular DNA with a length of 161,394 bp (Figure 1 and Table 1). The total lengths of the LSC, SSC and IRs are 90,628, 19,036 and 51,730 bp, respectively, and the sequences encode 134 genes, including eight rRNA genes, 40 tRNA genes and 86 protein-coding genes (Table A1). Different sections exhibit different distributions of genes: eight rRNA genes, 14 tRNA genes and 13 protein-coding genes within IR regions; one tRNA gene and 12 protein-coding genes in the SSC region; and 25 tRNA genes and 61 protein-coding genes within the LSC region (Figure 1 and Table 1). Furthermore, the GC contents of the entire cp genome and the IR, SSC and LSC regions are 36.80%, 42.70%, 30.90% and 34.60%, respectively, which are equivalent to the values obtained for other species in this study (Tables 1 and A1).

The results of the codon usage analysis are summarized in Table A2. Overall, these identified genes consist of 26,801 codons, and the most and least frequent amino acids in these codons are leucine (2828, 10.55%) and cysteine (308, 1.15%), respectively. The majority of the codons end in A- and U-.

The statistics of exons and introns are provided in Tables A3 and A4. The sequence contains 23 intron-containing genes, including *clpP* and *ycf3*, comprising two introns; in addition, ten of these intron-containing genes are located in LSC regions, and only *ndhA* is found in the SSC region. The longest intron (2511 bp) is found in *turnK-UUU*, and the smallest intron (483 bp) is located in *trnL-UAA*.

### 3.2. Analysis of Long Repeats and SSRs

The long-repeat analysis of the *Q. bawanglingensis* cp genome revealed that the genome contains eight more forward long repeats than inverted long repeats (21) (Table A5). The majority of the repeats are located in the LSC region (40), followed by the SSC region (12) and IRs (8). Moreover, a large proportion of repeats are located in intergenic regions (34, 68%), most of which are distributed in the LCS region, and the minority are found in the *trnS-GCU*, *trnS-UGA*, *trnG-GCC* (exon), *trnG-GCC*, *psaB*,

*psaA*, *clpP*, *rpl2*, *ndhF*, *ndhI*, *ndhA* (intron), *ycf1*, *trnV-UAC*, *trnA-UGC* and *rpl2* genes. Significantly, a longer repeat was not found in the *Q. bawanglingensis* cp genome, whose repeats range from 18 to 31 bp.

Based on the SSR polymorphism results, we found 82 SSRs in the *Q. bawanglingensis* cp genome. Most of the SSRs are distributed in the LSC region (62, 75.61%), followed by the SSC region (16, 19.51%) and IRs (4, 4.88%), whereas 64 are located in intergenic spaces and 18 in genes, such as *trnK-UUU*, *trnG-GC*, *atpF*, *rpoC2*, *rpoC1*, *rpoB*, *atpB*, *accD*, *clpP*, *petB*, *petD*, *ndhF*, *ndhD*, *ndhA* and *ycf1* (Table A6). Furthermore, *rpoC1* and *rpoC2* contain more SSR loci than the other genes. The cpSSRs in the cp genome generally consist of 69 mononucleotide SSRs (poly A or poly T), six dinucleotide SSRs and seven trinucleotide SSRs.

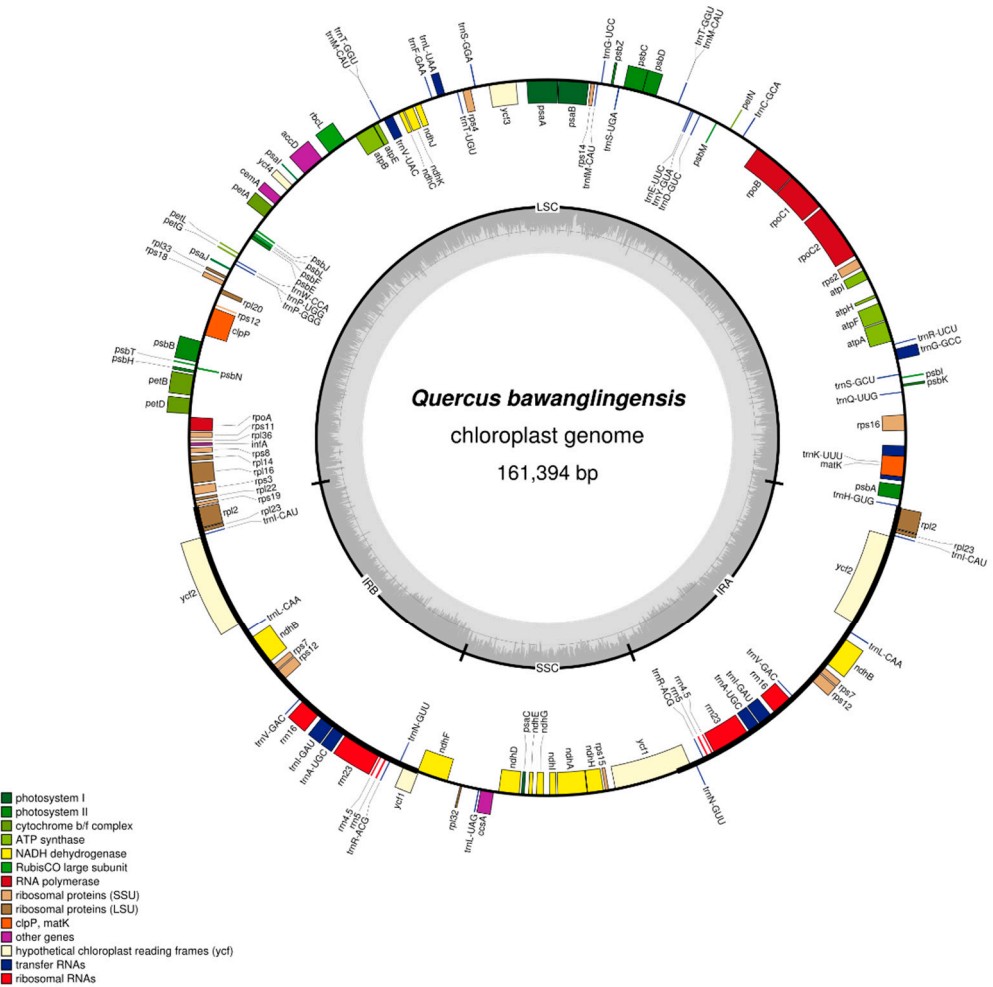

**Figure 1.** Map of the chloroplast genome of *Q. bawanglingensis*. The genes in the clockwise direction fill the inner circle, and the outer circle contains genes in the counterclockwise direction. Different colours represent different genes in different functional groups. The lighter grey shows the A + T content, and the darker grey in the inner circle indicates the G + C content. The direction of the genes is denoted by the direction of the grey arrow.

**Table 1.** Comparison of features of nine Fagaceae chloroplast genomes.

| Genome Features | Genome Size (bp) | LSC Length (bp) | SSC Length (bp) | IRs Length (bp) | Number of Genes | Number of Protein Coding Genes | Number of tRNA Genes | Number of rRNA Genes | GC Content (%) |
|---|---|---|---|---|---|---|---|---|---|
| *Q. bawanglingensis* Huang, Li et Xing | 161,394 | 90,628 | 19,036 | 51,730 | 134 | 86 | 40 | 8 | 36.8 |
| *Q. tarokoensis* Hayata | 161,355 | 90,602 | 19,033 | 51,720 | 134 | 86 | 40 | 8 | 36.9 |
| *Q. aliena var. acutiserrata* Maxim. ex Wenz. | 161,153 | 90,457 | 19,044 | 51,652 | 134 | 86 | 40 | 8 | 36.8 |
| *Q. variabilis* Bl. | 161,077 | 90,387 | 19,056 | 51,634 | 134 | 86 | 40 | 8 | 36.8 |
| *Q. baronii* Skan | 161,072 | 90,341 | 19,045 | 51,686 | 134 | 86 | 40 | 8 | 36.8 |
| *Q. aquifolioides* Rehd. et Wils. | 161,225 | 90,535 | 19,000 | 51,690 | 134 | 86 | 40 | 8 | 36.8 |
| *Fagus engleriana* Seem. | 158,346 | 87,667 | 18,895 | 51,784 | 131 | 83 | 40 | 8 | 37.1 |
| *Lithocarpus balansae* (Drake) A. Camus | 161,020 | 90,596 | 19,160 | 51,264 | 134 | 87 | 39 | 8 | 36.7 |
| *Castanea mollissima* Bl. | 160,799 | 90,432 | 18,995 | 51,372 | 130 | 83 | 37 | 8 | 36.8 |

SSC, a small single-copy region; LSC, a large single-copy region; IRs, two inverted repeats.

### 3.3. Comparison of Complete Chloroplast Genomes among Fagaceae Species

We performed a Blast analysis of the sequences from nine cp genomes using mVISTA, and the cp genome of *Q. tarokoensis* was used as the reference (Figure 2). The results showed that the entire genome is well conserved across all species with the exception of *F. engleriana*. The SCs have a substantially higher nucleotide diversity than the IRs, whereas more variation was found in the noncoding regions compared with the coding regions, consistent with the observations of the nucleotide variability (pi), which showed that the pi values of LSC, SSC and IRb are 0.004906, 0.007103 and 0.000729, respectively, among the six species (*Q. bawanglingensis*, *Q. variabilis*, *Q. aliena var. acutiserrata*, *Q. aquifolioides*, *Q. baronii*, and *Q. tarokoensis*); this information is graphically presented in Figure 3. Importantly, both the results from the mVISTA analysis and the assessment of nucleotide variability showed that numerous divergence hotspot regions, such as *rbcL-accD* (pi: 0.02365, 0.02317), *accD* (pi: 0.02365), *trnS-trnG* (pi: 0.01865, 0.01802), *ycf1* (pi: 0.01643, 0.01627), *trnG-trnR* (pi: 0.0173), *trnK-rps16* (pi: 0.01627), *ndhF* (pi: 0.01619) and *trnH-psbA* (pi:0.01548), are completely located within the SC regions (Figures 2–4). In addition, more variable sites are located in intergenic regions than in coding genes, which allows the potential development of DNA barcodes for species identification and taxonomical studies of the genus *Quercus*.

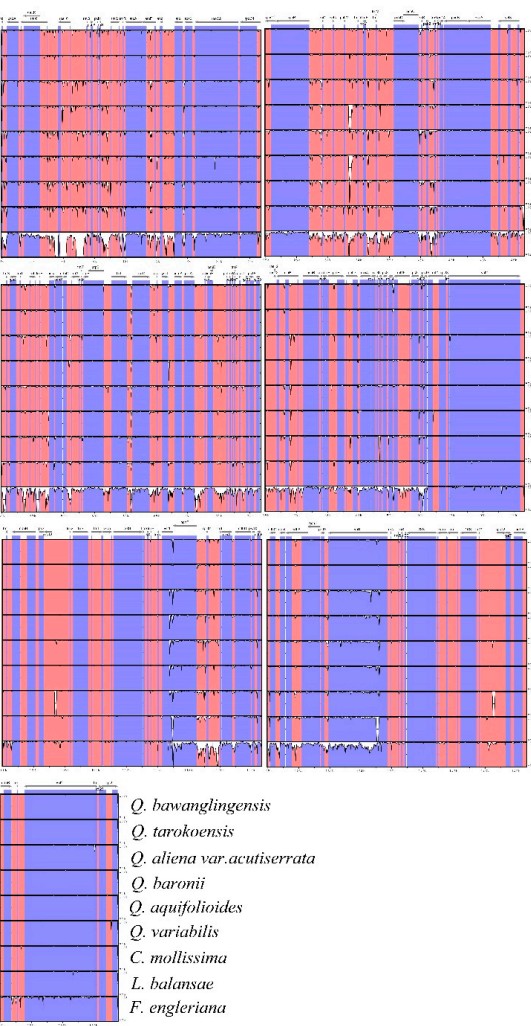

**Figure 2.** Sequence identity plots of the nine Fagaceae cp genomes generated by mVISTA, with the *Q. tarokoensis* genome as the reference. The vertical and horizontal axes in the figure represent the consistency degree of the sequences from 50% to 100% and the sequence length, respectively. Annotated genes are displayed along the top.

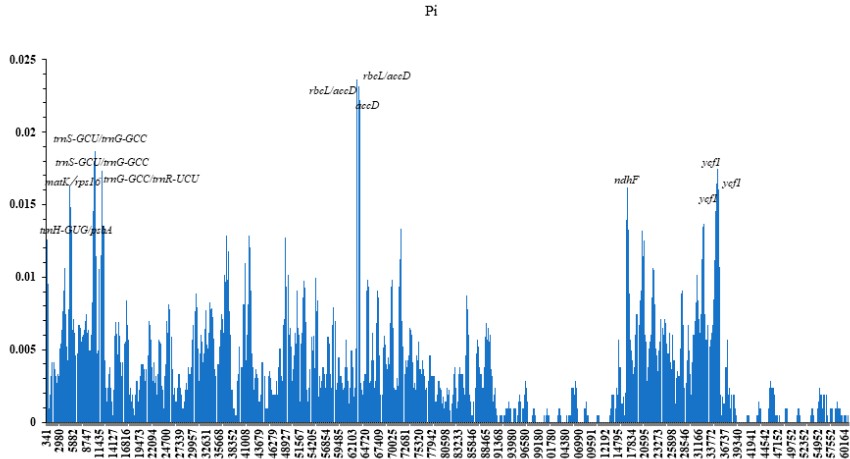

**Figure 3.** Nucleotide variability (pi) values. *X*-axis: position of the midpoint of a window. *Y*-axis: nucleotide diversity of each window.

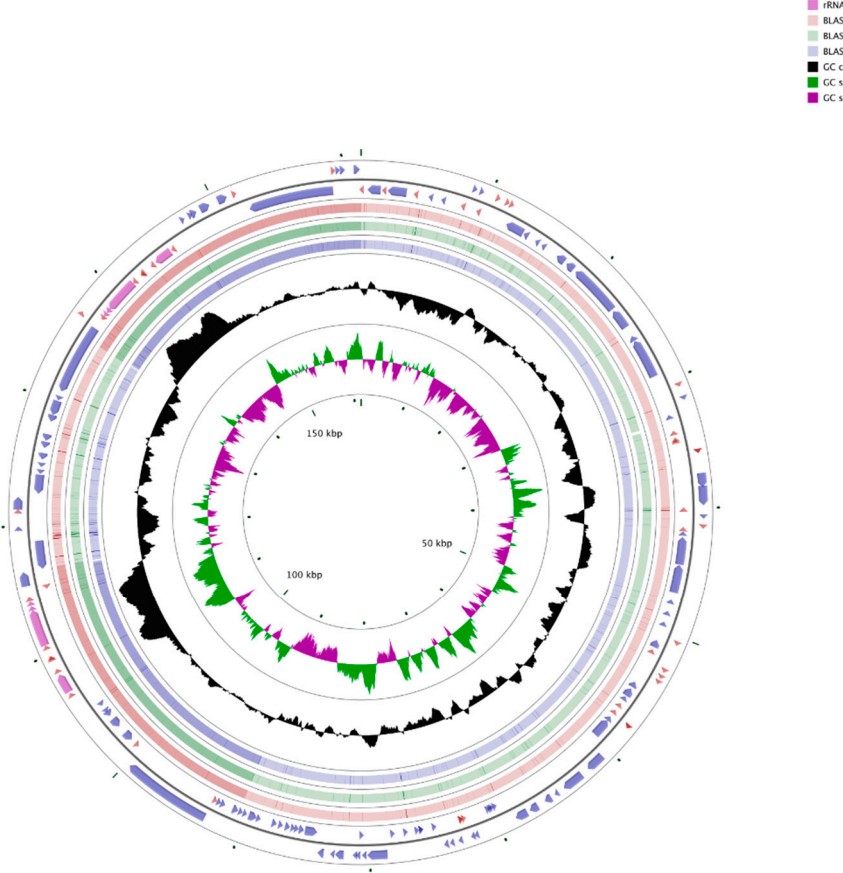

**Figure 4.** Comparison of the cp genomes from four Fagaceae species. The outer two rings pointing in different directions show the coding sequence (CDS), rRNA genes, and tRNA genes. The three inner circles show the blast results for *Q. bawanglingensis* vs. *L. balansae*, *Quercus tarokoensis* and *Q. variabilis*, respectively. GC skew+ (in a green colour) means G > C, whereas GC skew- (in a purple colour) indicates G < C.

The IRs are extremely conserved in *Quercus* (Figure 5), consistent with the observations shown in Figures 2–4 but are slightly different from the others investigated in this study. The *rps19* gene is located within the LSC, 10 bp from the border of the LSC/IRb, in all species with the exception

of *C. mollissima* (0 bp), and this gene is also found 16 bp between the *trnH* gene in the LSC and the IRA/LSC border in all species except *C. mollissima*, in which the gene is found at a distance of 8 bp. At the boundary of the LSC/IRb, the *rpl2* gene is located 62 bp from the LSC, whereas shorter distances were found in *C. mollissima* (67 bp) and *F. engleriana* (65 bp). In *Quercus* species, the boundary of the LSC/IRs is highly conserved, whereas the borders of IRs/SSC are highly variable. The IRs/SSC borders are generally located in the varied sites of the *ycf1* and *ndhF* genes. The junctions of SSC/IRa located in *ycf1* within the SSC and IRa regions vary in length (*Q. bawanglingensis*: 4653 and 1038 bp; *Q. tarokoensis*: 4625 and 1064 bp; *Q. aliena var. acutiserrata*: 4615 and 1043 bp; *Q. variabilis*: 4620 and 1041 bp; *Q. baronii*: 4611 and 1047 bp; *Q. aquifolioides*: 4513 and 1057 bp; *C. mollissima*: 4623 and 1059 bp; *L. balansae*: 4626 and 828 bp; and *F. engleriana*: 4633 and 1049 bp). The *ndhF* gene relevant for photosynthesis was found to be located at 1 to 159 bp from the IRb/SSC junction.

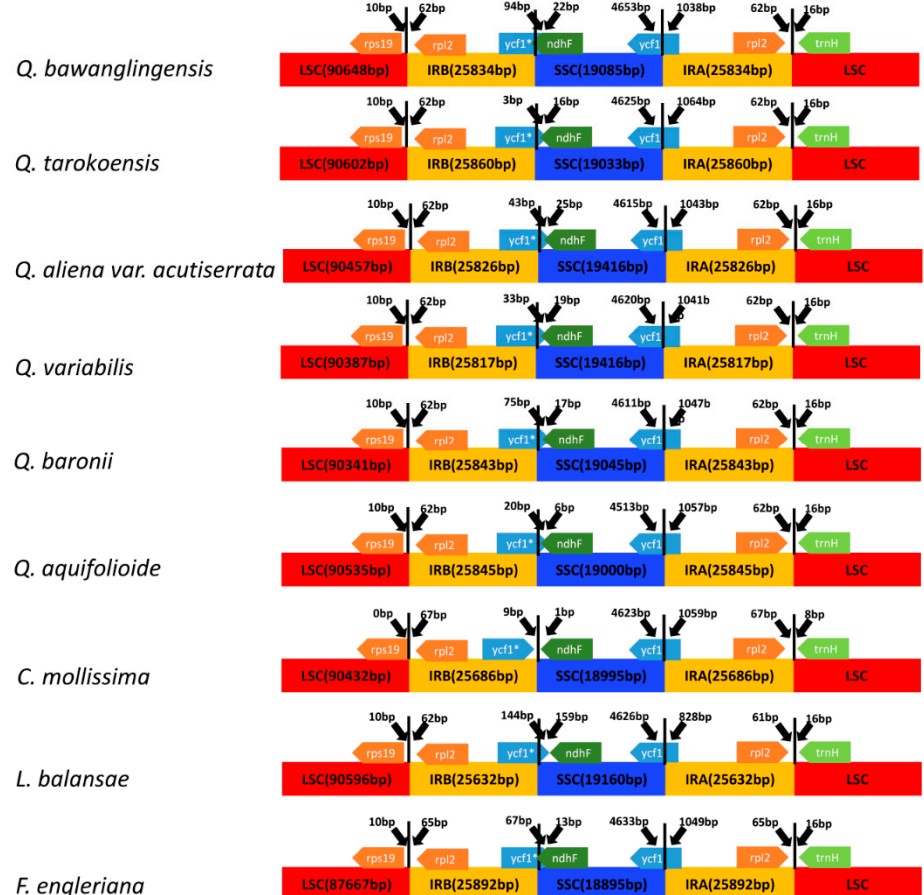

**Figure 5.** Comparison of the borders for the LSC and SSC regions and IRs among the nine Fagaceae cp genomes.

### 3.4. Phylogenetic Analysis

With *P. trichocarpa* and *T. cacao* as the outgroups, a phylogenetic tree was generated using ML based on the above-described whole-cp genome data (Figure 6). The phylogenetic results resolved 29 nodes with bootstrap support values of 52–100, which generally strongly supports the hypothesis that the Fagaceae species form a single clade. *F. engleriana* is located at the top node as a sister to *T. doichangensis* with high support, whereas *L. balansae* and *Castanopsis* species closely related to group *Cyclobalanopsis* are split into *Quercus*. The clade formed by *Quercus* indubitably involves group *Quercus*, whereas group *Ilex* is both separately clustered with group *Cyclobalanopsis*, and *Cerris*. *Q. bawanglingensis* is located in one clade that includes several evergreen oaks. The phylogenetic tree also revealed that *Q. bawanglingensis* is a sister to *Q. tarokoensis* with a 100% bootstrap value.

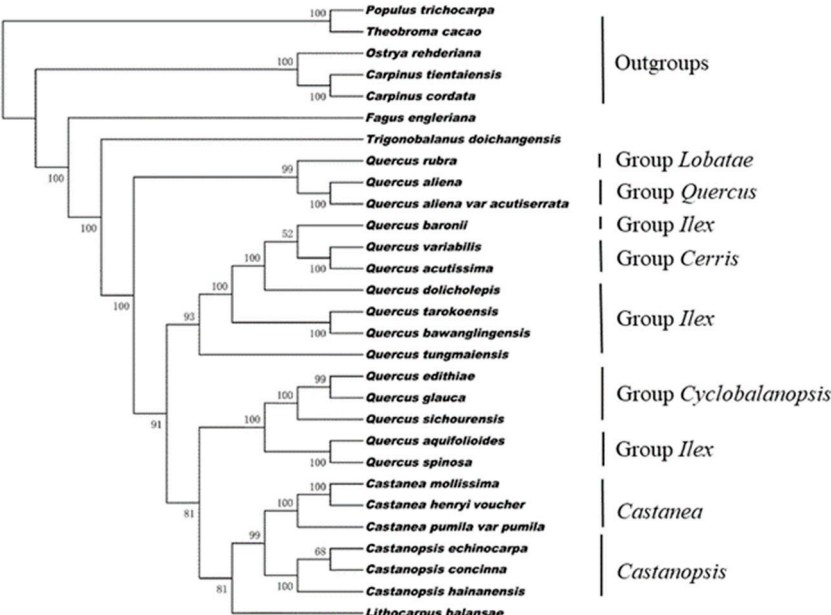

**Figure 6.** Maximum likelihood (ML) phylogenetic tree of 29 species of Fagaceae constructed using their chloroplast genomes. *Populus trichocarpa* and *Theobroma cacao* were used as the outgroups.

## 4. Discussion

In general, the complete cp genome of *Q. bawanglingensis* has a strong resemblance to those of other *Quercus* species in the aspects of genome size and structure, GC content, genes and gene order, which illustrates that the cp genomes are conserved in *Quercus* [43–45,62,63]. Nonetheless, changes in the border of LSC/IRb and the nucleotide variability were detected, which are relatively common in plants [15,46,64]. The maximum difference in genome size among the nine Fagaceae species is 3055 bp, whereas the largest difference in the LSC region is 2981 bp, which could indicate that the divergence in the LSC length leads to variation in the size of the cp genomes based on IR contraction or expansion [31]. Differences in the four IR boundaries among species frequently appear during the process of cp genome evolution, which leads to further changes in the cp genome size. Hence, IR regions are used to explain size differences between cp genomes due to their contraction and expansion at the borders, even though they are the most conserved regions in cp genome sequences [64–67].

Higher nucleotide diversity has been found in SCs compared with IRs and in noncoding regions compared with coding regions, which is in accordance with the results found for other taxa [43–45,63,68], although exceptions have been identified [32,69]. A cp genome has a copy-dependent repair mechanism that ensures the uniformity and stability of two IR regions in sequence and enhances the stability and conservation of the genome [70,71], which might explain the lower sequence divergence in the IRs compared with the LSC or SSC regions, because natural selection coding regions are more conserved than non-coding regions [72]. In our study, both the results from the mVISTA analysis and the nucleotide variability (pi) assessment showed that numerous divergence hotspot regions are primarily situated in the SCs of the cp genome and that more variable sites are located in intergenic regions than in coding genes, and these can be directly utilized for the development of new molecular markers for research on *Quercus* species identification and taxonomy. Among these divergence hotspot regions, *trnH-psbA* has already been selected as a suitable barcode for plants [40,73], as have *rbcL-accD*, *trnS-trnG* [74], *ndhF* [40,75], *ycf1* [69,76], *accD* [67,77], *trnG-trnR* [78] and *trnK-rps16* [79]. In this study, the *ycf1*, *ndhF* and *accD* genes were found to be optimal genetic markers based on their high substitution variability, repeat sequence diversity and SSC/IR junction length variability. The *accD* gene, which encodes the acetyl-CoA carboxylase (ACCase) enzyme, is crucial for maintenance of the plastid compartment and for leaf development in tobacco [67] and might be considered a locus for obtaining insights into chloroplast genome evolution [77] in *Quercus*. As a NADH dehydrogenase

gene, *ndhF* is favoured by studies on the evolution of plant taxonomy [79–81]. The *ycf1* gene, which has the largest open reading frame, is crucial for the protein translocons at the inner envelope membranes in Chloroplasts (TIC) complex, which related to plant survival, due to Tic214/Tic20, which provides access of cps to exotic proteins [82]. The *ycf1* gene is also important for examining the diversification of the cp genome in algae or other plants [83]. Further research is necessary for examining whether these divergence hotspot regions could be used for assessing the taxonomic evolution of *Quercus* or could be considered candidate DNA barcodes.

The observed GC content is generally consistent with the results of previous intensive studies [3,84–87], which confirms that the cp genome of Fagaceae species is rich in adenine and thymine (AT). GC skewness is considered an indicator of replication terminals, replication origin, lag chains and DNA lead chains [88–90] as well as a dominant factor in codon bias. Several studies [3,84,87,91] have suggested that high AT richness is the major reason for synonymous codons ending in A/U. This phenomenon might be subordinate to natural selection and mutation during the process of evolution.

cpSSRs, which are typical uniparentally inherited material, have been used extensively in analyses of taxonomic status, phylogenetic relationships, the maternal structure of the community, diversity and differentiation [92–94]. SSR polymorphisms result from a mutational mechanism in which SSRs with a length of at least 10 bp appear as slipped-strand mispairings [95]. We found 82 SSRs in the *Q. bawanglingensis* cp genome that were mostly distributed in the LSC (62, 75.61%) and intergenic spaces (64, 78.04%). Efficient molecular markers might be selected by using auxiliary information from the uneven distribution of cpSSRs for phylogenetic and phylogeographical studies [38,96,97]. In addition, the majority of cpSSRs in the *Q. bawanglingensis* cp genome mononucleotides and dinucleotides are formed by A and T, which might be related to the high AT richness in the nucleotide composition, similar to the results found for other cp genomes [39,43,46,98].

Previous studies on the origin time of Fagaceae have shown that fossils of *T. doichangensis* were the first to appear in the fossil record and that *Cyclobalanopsis* is closer than *Quercus* to the ancestral group in Fagaceae [99]. Based on the phylogenetic trees, *F. engleriana* and *T. doichangensis* are located in the basal phylogeny, and the evolutionary tree is consistent with the fossil record [99]. *Q. bawanglingensis* and *Q. tarokoensis* have a close relationship, and in accordance with their morphological features, both of these species belong to section *Engleriana* in group *Ilex* [73,100]. Importantly, the *Quercus* species were not shown to form a clade, similar to the findings in other research [44–46]. Group *Ilex* within group *Cerris* forms a *Cerris-Ilex* clade, which is identified by inferences from primarily chloroplast haplotypes between group *Cerris* and its sister group, *Ilex* [39]. A group comprising *Heterobalanus* (corresponding to group *Ilex*) and *Cyclobalanopsis* matches the traditional taxonomy, which formalized both *Cyclobalanopsis* and *Ilex* as one lineage [101]. Overall, the relationships among the other branches in Fagaceae are mostly consistent with those inferred from nuclear data [33,102].

## 5. Conclusions

In the present study, we successfully completed the whole cp genome for the vulnerable oak tree *Q. bawanglingensis* using next generation sequencing technology. In comparing the *Q. bawanglingensis* cp genome with prior *Quercue* species from NCBI, we found that it was very similar in cp genome structure and gene content. Nevertheless, obviously heterogeneous sequence divergences were revealed in different regions among *Quercus* cp genomes. The divergence hotspot regions and abundant SSRs identified in the cp genome could be used for molecular marker development for further population genetics studies on whether and how natural populations have adapted to their local environments, to predict their responses to future habitat alterations and to establish adequate conservation strategies for this vulnerable species. The phylogenetic relationships of *Q. bawanglingensis* in Fagaceae were robustly resolved based on the cp genome data, strongly supporting the sister relationship between *Q. bawanglingensis* and *Q. tarokoensis* in the group *Ilex* lineage. Overall, the data obtained will contribute to further studies on the diversity, ecology, taxonomy, phylogenetic evolution and conservation of Chinese *Quercus* species.

**Author Contributions:** The experiments were conceived and designed by Z.-P.J. and J.-F.L.; X.L., E.-M.C., Y.-N.H., Y.W., and N.Y. were involved in the collection of the study materials. X.L. and E.-M.C. participated in the DNA extraction and data analyses. X.L. wrote and J.-F.L. revised the manuscript. All authors read and approved the final manuscript.

**Funding:** This study was funded by the Fundamental Research Funds for the Central Non-Profit Research Institution of CAF (CAFYBB2018ZB001).

**Acknowledgments:** The authors sincerely thank Mingzhi Li of Genepioneer Biotechnologies Co. Ltd., Nanjing, China for the assistance provided with this study. In addition, the authors sincerely thank the reviewers for their careful reading and helpful comments on this manuscript.

**Conflicts of Interest:** The authors declare no conflict of interest.

**Appendix A**

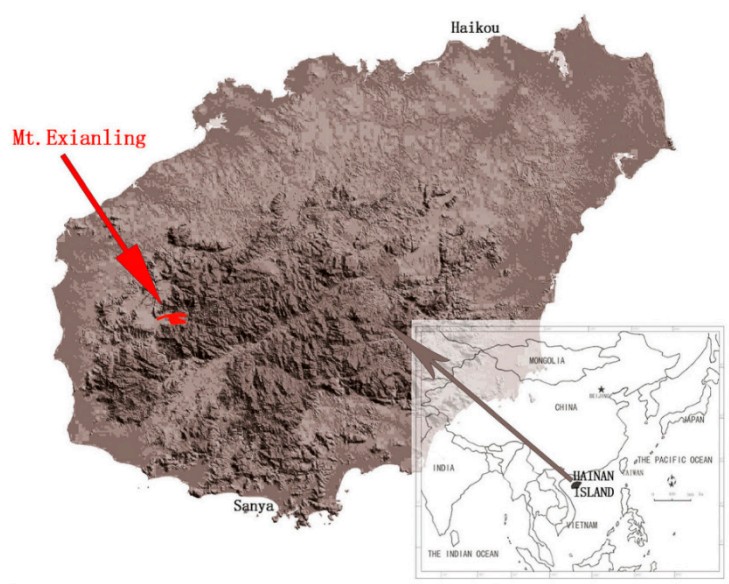

**Figure A1.** Location of Mt. Exianling, Hainan Island, China.

**Table A1.** Base composition of the *Q. bawanglingensis* chloroplast genome.

| Region | CDS | tRNA Genes | rRNA Genes | A (%) | T (U) (%) | C (%) | G (%) | G + C (%) |
|--------|-----|-----------|-----------|-------|-----------|-------|-------|-----------|
| LSC | 61 | 25 | | 32.00 | 33.40 | 17.70 | 16.90 | 34.60 |
| SSC | 12 | 1 | | 34.40 | 34.70 | 16.30 | 14.60 | 30.90 |
| IRs | 13 | 14 | 8 | 28.60 | 28.60 | 21.40 | 21.40 | 42.70 |
| Total | 86 | 40 | 8 | 31.20 | 32.00 | 18.70 | 18.10 | 36.80 |

**Table A2.** Detailed statistics on codon usage, diversification in synonymous codon usage, relative synonymous codon usage (RSCU) values and codon-anticodon recognition patterns of the *Q. bawanglingensis* chloroplast genome.

| Amino Acid | Codon | No. | RSCU | tRNA | Amino Acid | Codon | No. | RSCU | tRNA |
|------------|-------|-----|------|------|------------|-------|-----|------|------|
| **Phe** | UUU | 986 | 1.29 | | Glu | GAG | 354 | 0.5 | |
| **Phe** | UUC | 537 | 0.71 | trnF-GAA | Ser | UCU | 559 | 1.66 | |
| **Leu** | UUA | 896 | 1.9 | trnL-UAA | Ser | UCC | 350 | 1.04 | Trns-GGA |
| **Leu** | UUG | 570 | 1.21 | trnL-CAA | Ser | UCA | 404 | 1.20 | trnS-UGA |
| **Leu** | CUU | 590 | 1.25 | | Ser | UCG | 192 | 0.57 | |
| **Leu** | CUC | 203 | 0.43 | | Ser | AGU | 394 | 1.17 | |
| **Leu** | CUA | 373 | 0.79 | trnL-UAG | Ser | AGC | 127 | 0.38 | trnS-GCU |
| **Leu** | CUG | 196 | 0.42 | | Pro | CCU | 408 | 1.47 | |

**Table A2.** *Cont.*

| Amino Acid | Codon | No. | RSCU | tRNA | Amino Acid | Codon | No. | RSCU | tRNA |
|---|---|---|---|---|---|---|---|---|---|
| **Ile** | AUU | 1137 | 1.45 | | Pro | CCC | 225 | 0.81 | trnP-GGG |
| **Ile** | AUC | 455 | 0.58 | trnI-GAU | Pro | CCA | 313 | 1.13 | trnP-UGG |
| **Ile** | AUA | 758 | 0.97 | | Pro | CCG | 163 | 0.59 | |
| **Met** | AUG | 618 | 1.00 | trnM-CAU trnI-CAU | Thr | ACU | 535 | 1.59 | |
| **Val** | GUU | 509 | 1.42 | | Thr | ACC | 247 | 0.73 | |
| **Val** | GUC | 180 | 0.50 | trnV-GAC | Thr | ACA | 403 | 1.20 | trnT-UGU |
| **Val** | GUA | 545 | 1.52 | trnV-UAC | Thr | ACG | 162 | 0.48 | |
| **Val** | GUG | 204 | 0.57 | | Ala | GCU | 632 | 1.80 | |
| **Tyr** | UAU | 798 | 1.58 | | Ala | GCC | 221 | 0.63 | |
| **Tyr** | UAC | 212 | 0.42 | trnY-GUA | Ala | GCA | 384 | 1.09 | trnA-UGC |
| **Ter** | UAA | 47 | 1.64 | | Ala | GCG | 169 | 0.48 | |
| **Ter** | UAG | 21 | 0.73 | | Cys | UGU | 222 | 1.44 | |
| **Ter** | UGA | 18 | 0.63 | | Cys | UGC | 86 | 0.56 | trnC-GCA |
| **His** | CAU | 486 | 1.54 | | Try | UGG | 463 | 1.00 | trnW-CCA |
| **His** | CAC | 146 | 0.46 | trnH-GUG | Arg | CGU | 336 | 1.25 | |
| **Gln** | CAA | 735 | 1.55 | trnQ-UUG | Arg | CGC | 109 | 0.41 | |
| **Gln** | CAG | 215 | 0.45 | | Arg | CGA | 354 | 1.32 | trnR-ACG |
| **Asn** | AAU | 1012 | 1.54 | | Arg | CGG | 122 | 0.45 | |
| **Asn** | AAC | 302 | 0.46 | | Arg | AGA | 505 | 1.88 | trnR-UCU |
| **Lys** | AAA | 1070 | 1.47 | | Arg | AGG | 186 | 0.69 | |
| **Lys** | AAC | 383 | 0.53 | trnN-GUU | Gly | GGU | 582 | 1.28 | |
| **Asp** | GAU | 872 | 1.61 | | Gly | GGC | 208 | 0.46 | trnG-GCC |
| **Asp** | GAC | 208 | 0.39 | trnD-GUC | Gly | GGA | 707 | 1.55 | trnG-CCC |
| **Glu** | GAA | 1069 | 1.5 | trnE-UUC | Gly | GGG | 328 | 0.72 | |

**Table A3.** List of annotated genes in the *Q. bawanglingensis* chloroplast genome.

| Category for Genes | Group of Gene | Name of Gene |
|---|---|---|
| | Photosystem I | *psaA, psaB, psaC, psaI, psaJ,* |
| | Photosystem II | *psbA, psbB, psbC, psbD, psbE, psbF, psbH, psbI, psbJ, psbK, psbL, psbM, psbN, psbT, psbZ* |
| **Photosynthesis related genes** | Cytochrome b/f complex | *petA, petB[1], petD[1], petL, petG, petN* |
| | ATP synthase | *atpA, atpB, atpE, atpF[1], atPH, atpI* |
| | Cytochrome c synthesis | *ccsA* |
| | Assembly/stability of photosystem | *ycf3[2], ycf4* |
| | NADPH dehydrogenase | *ndhA[1], ndhB[1d], ndhC, ndhD, ndhE, ndhF, ndhG, ndhH, ndhI, ndhJ, ndhK* |
| | Rubisco | *rbcL* |
| **Transcription and translation related genes** | Transcription | *rpoC1[1], rpoC2, rpoA, rpoB* |
| | Ribosomal proteins | *rps2, rps3, rps4, rps7[d], rps8, rps11, rps12[d], rps14, rps15, rps16[1], rps18, rps19,* |
| | Large subunit | *rpl2[1], rpl14, rpl16[1], rpl20, rpl22, rpl23[d], rpl32, rpl33, rpl36* |
| **RNA genes** | Ribosomal RNA | *4.5S rRNA[d], 5S rRNA[d], 16S rRNA[d], 23S rRNA[d]* |
| | Transfer RNA | *trnH-GUG, trnK-UUU[1], trnQ-UUG, trnS-GCU, trnG-GCC[1], trnR-UCU, trnC-GCA, trnD-GUC, trnY-GUA, trnE-UUC, trnT-GGU[d], trnM-CAU, trnS-UGA, trnG-UCC, trnfM-CAU, trnS-GGA, trnT-UGU, trnL-UAA[1], trnF-GAA, trnV-UAC[1], trnW-CCA, trnP-UGG, trnP-GGG, trnL-CAA[d], trnV-GAC[d], trnI-GAU[1d], trnR-ACG[d], trnL-UAG, trnN-GUU[d], trnA-UGC[1d], trnI-CAU[d]* |
| | RNA processing | *matK* |
| | Carbon metabolism | *cemA* |
| **Other genes** | Fatty acid synthesis | *accD* |
| | Proteolysis | *clpP[2]* |
| | Translational initiation factor | *infA* |
| **Genes of unknown function** | Conserved reading frames | *ycf1[d], ycf2[d]* |

[1], genes containing only one intron; [2], genes containing two introns; [d], two gene copies in the IRs.

**Table A4.** The lengths of introns and exons in genes in the *Q. bawanglingensis* chloroplast genome.

| Gene | Strands | Location | Exon1 (bp) | Exon2 (bp) | Intron1 (bp) | Exon3 (bp) | Intron2 (bp) |
|------|---------|----------|-----------|-----------|--------------|-----------|--------------|
| *trnK-UUU* | – | LSC | 37 | 35 | 2511 | | |
| *trnI-GAU* | + | IRA | 37 | 35 | 955 | | |
| *trnI-GAU* | - | IRB | 42 | 35 | 950 | | |
| *trnA-UGC* | + | IRA | 38 | 35 | 800 | | |
| *trnA-UGC* | | IRB | 38 | 35 | 800 | | |
| *trnG-GCC* | + | IRA | 23 | 37 | 736 | | |
| *trnV-UAC* | - | LSC | 38 | 35 | 630 | | |
| *trnL-UAA* | + | LSC | 35 | 50 | 483 | | |
| *rps12* | + | IRB | | 232 | 536 | 26 | |
| *rps12* | - | IRA | | 231 | 537 | 30 | |
| *rpoC1* | - | LSC | 432 | 1626 | 833 | | |
| *ndhB* | + | IRB | 777 | 756 | 680 | | |
| *ndhB* | - | IRA | 777 | 756 | 680 | | |
| *ndhA* | - | SSC | 552 | 540 | 1037 | | |
| *clpP* | - | LSC | 69 | 294 | 862 | 228 | 653 |
| *ycf3* | - | LSC | 126 | 228 | 721 | 153 | 768 |
| *rpl16* | - | LSC | 9 | 399 | 1102 | | |
| *rpl2* | - | IRA | 390 | 471 | 648 | | |
| *rpl2* | + | IRB | 393 | 471 | 645 | | |
| *petB* | + | LSC | 6 | 642 | 843 | | |
| *atpF* | - | IRB | 144 | 411 | 770 | | |
| *rps16* | - | LSC | 42 | 228 | 899 | | |
| *petD* | + | LSC | 9 | 474 | 640 | | |

**Table A5.** Repeated sequences of the *Q. bawanglingensis* chloroplast genome.

| ID | Size (bp) | Repeat Start I | Type | Size (bp) | Repeat Start 2 | Mismatch (bp) | E-Value | Region | Gene |
|----|-----------|----------------|------|-----------|----------------|---------------|---------|--------|------|
| 1 | 18 | 325 | F | 18 | 4926 | 0 | $1.07 \times 10^{-1}$ | LSC | |
| 2 | 21 | 6821 | R | 21 | 6821 | 0 | $1.67 \times 10^{-3}$ | LSC | |
| 3 | 19 | 6835 | R | 19 | 6835 | 0 | $2.67 \times 10^{-2}$ | LSC | |
| 4 | 18 | 7431 | R | 18 | 7431 | 0 | $1.07 \times 10^{-1}$ | LSC | |
| 5 | 18 | 8884 | R | 18 | 8884 | 0 | $1.07 \times 10^{-1}$ | LSC | |
| 6 | 18 | 9988 | R | 18 | 9988 | 0 | $1.07 \times 10^{-1}$ | LSC | |
| 7 | 31 | 11,852 | R | 31 | 11,852 | 0 | $1.59 \times 10^{-9}$ | LSC | |
| 8 | 22 | 30,370 | F | 22 | 30,388 | 0 | $4.16 \times 10^{-4}$ | LSC | |
| 9 | 20 | 10,290 | F | 20 | 31,747 | 0 | $6.66 \times 10^{-3}$ | LSC | |
| 10 | 19 | 8557 | R | 19 | 35,014 | 0 | $2.67 \times 10^{-2}$ | LSC | |
| 11 | 20 | 4925 | F | 20 | 36,722 | 0 | $6.66 \times 10^{-3}$ | LSC | |
| 12 | 18 | 325 | F | 18 | 36,723 | 0 | $1.07 \times 10^{-1}$ | LSC | |
| 13 | 21 | 9531 | F | 21 | 40,098 | 0 | $1.67 \times 10^{-3}$ | LSC | *trnS-GCU, trnS-UGA* |
| 14 | 20 | 40,206 | R | 20 | 40,206 | 0 | $6.66 \times 10^{-3}$ | LSC | |
| 15 | 22 | 11,376 | F | 22 | 41,438 | 0 | $4.16 \times 10^{-4}$ | LSC | *trnG-GCC* (exon), *trnG-GCC* |
| 16 | 18 | 43,688 | F | 18 | 45,912 | 0 | $1.07 \times 10^{-1}$ | LSC | *psaB, psaA* |
| 17 | 19 | 21,298 | R | 19 | 54,384 | 0 | $2.67 \times 10^{-2}$ | LSC | |
| 18 | 21 | 54,575 | F | 21 | 54,594 | 0 | $1.67 \times 10^{-3}$ | LSC | |
| 19 | 20 | 56,125 | R | 20 | 56,125 | 0 | $6.66 \times 10^{-3}$ | LSC | *ndhC* |
| 20 | 21 | 62,263 | R | 21 | 62,263 | 0 | $1.67 \times 10^{-3}$ | LSC | |
| 21 | 19 | 64,976 | R | 19 | 64,976 | 0 | $2.67 \times 10^{-2}$ | LSC | |
| 22 | 21 | 69,245 | R | 21 | 69,245 | 0 | $1.67 \times 10^{-3}$ | LSC | |
| 23 | 18 | 69,246 | R | 18 | 69,246 | 0 | $1.07 \times 10^{-1}$ | LSC | |
| 24 | 18 | 69,246 | F | 18 | 69,247 | 0 | $1.07 \times 10^{-1}$ | LSC | |
| 25 | 18 | 69,247 | R | 18 | 69,247 | 0 | $1.07 \times 10^{-1}$ | LSC | |
| 26 | 19 | 71,499 | R | 19 | 71,499 | 0 | $2.67 \times 10^{-2}$ | LSC | |
| 27 | 19 | 72,775 | R | 19 | 72,775 | 0 | $2.67 \times 10^{-2}$ | LSC | |
| 28 | 18 | 18,660 | F | 18 | 76,843 | 0 | $1.07 \times 10^{-1}$ | LSC | *clpP* |
| 29 | 18 | 52,390 | F | 18 | 87,369 | 0 | $1.07 \times 10^{-1}$ | LSC | |
| 30 | 20 | 91,234 | F | 20 | 91,254 | 0 | $6.66 \times 10^{-3}$ | IRB | *rpl2* |
| 31 | 20 | 105,557 | F | 20 | 105,575 | 0 | $6.66 \times 10^{-3}$ | IRB | |
| 32 | 23 | 113,771 | F | 23 | 113,802 | 0 | $1.04 \times 10^{-4}$ | IRB | |

**Table A5.** *Cont.*

| ID | Size (bp) | Repeat Start I | Type | Size (bp) | Repeat Start 2 | Mismatch (bp) | E-Value | Region | Gene |
|---|---|---|---|---|---|---|---|---|---|
| 33 | 18 | 69,461 | F | 18 | 116,760 | 0 | $1.07 \times 10^{-1}$ | LSC, SSC | *ndhF* |
| 34 | 21 | 117,268 | R | 21 | 117,268 | 0 | $1.67 \times 10^{-3}$ | SSC | *ndhF* |
| 35 | 18 | 66,388 | F | 18 | 118,801 | 0 | $1.07 \times 10^{-01}$ | LSC, SSC | |
| 36 | 18 | 4934 | F | 18 | 118,916 | 0 | $1.07 \times 10^{-1}$ | LSC, SSC | |
| 37 | 19 | 18,660 | R | 19 | 119,064 | 0 | $2.67 \times 10^{-2}$ | LSC, SSC | |
| 38 | 23 | 119,066 | R | 23 | 119,066 | 0 | $1.04 \times 10^{-4}$ | SSC | |
| 39 | 19 | 10,289 | F | 19 | 126,141 | 0 | $2.67 \times 10^{-2}$ | LSC, SSC | |
| 40 | 18 | 31,747 | F | 18 | 126,142 | 0 | $1.07 \times 10^{-1}$ | LSC, SSC | |
| 41 | 19 | 73,588 | F | 19 | 127,650 | 0 | $2.67 \times 10^{-2}$ | LSC, SSC | *ndhA* |
| 42 | 25 | 127,669 | F | 25 | 127,693 | 0 | $6.51 \times 10^{-6}$ | SSC | *ndhA* (intron) |
| 43 | 20 | 119,064 | R | 20 | 130,690 | 0 | $6.66 \times 10^{-3}$ | SSC | |
| 44 | 19 | 18,660 | F | 19 | 130,691 | 0 | $2.67 \times 10^{-2}$ | LSC, SSC | |
| 45 | 18 | 10,551 | F | 18 | 133,570 | 0 | $1.07 \times 10^{-1}$ | LSC, SSC | *ycf1* |
| 46 | 24 | 116,026 | F | 24 | 135,972 | 0 | $2.60 \times 10^{-5}$ | IRB, IRA | *ycf1* |
| 47 | 23 | 138,197 | F | 23 | 138,228 | 0 | $1.04 \times 10^{-4}$ | IRA | |
| 48 | 20 | 57,490 | F | 20 | 142,313 | 0 | $6.66 \times 10^{-3}$ | LSC, IRA | *trnV-UAC, trnA-UGC* |
| 49 | 20 | 146,427 | F | 20 | 146,445 | 0 | $6.66 \times 10^{-3}$ | IRA | |
| 50 | 20 | 160,748 | F | 20 | 160,768 | 0 | $6.66 \times 10^{-3}$ | IRA | *rpl2* |

**Table A6.** Simple sequence repeats (SSRs) in the *Q. bawanglingensis* chloroplast genome.

| ID | SSR | Size | Start | End | Region | Gene | ID | SSR | Size | Start | End | Region | Gene |
|---|---|---|---|---|---|---|---|---|---|---|---|---|---|
| 1 | (A)11 | 11 | 333 | 343 | LSC | | 42 | (T)10 | 10 | 59,813 | 59,822 | LSC | *atpB* |
| 2 | (A)10 | 10 | 1796 | 1805 | LSC | | 43 | (T)11 | 11 | 60,285 | 60,295 | LSC | |
| 3 | (T)15 | 15 | 4116 | 4130 | LSC | *trnK-UUU* | 44 | (AT)6 | 12 | 62,268 | 62,279 | LSC | |
| 4 | (C)12(A)11 | 23 | 4426 | 4448 | LSC | | 45 | (T)11 | 11 | 64,317 | 64,327 | LSC | *accD* |
| 5 | (T)13 | 13 | 4690 | 4702 | LSC | | 46 | (A)10 | 10 | 64,492 | 64,501 | LSC | |
| 6 | (A)11 | 11 | 4934 | 4944 | LSC | | 47 | (AT)7 | 14 | 64,795 | 64,808 | LSC | |
| 7 | (A)11 | 11 | 5134 | 5144 | LSC | | 48 | (T)11 | 11 | 65,170 | 65,180 | LSC | |
| 8 | (T)11 | 11 | 6967 | 6977 | LSC | | 49 | (T)10 | 10 | 66,211 | 66,220 | LSC | |
| 9 | (A)10 | 10 | 8139 | 8148 | LSC | | 50 | (T)14 | 14 | 66,389 | 66,402 | LSC | |
| 10 | (A)16 | 16 | 8555 | 8570 | LSC | | 51 | (T)10 | 10 | 68,836 | 68,845 | LSC | |
| 11 | (A)10 | 10 | 8889 | 8898 | LSC | | 52 | (A)19(AT)6 | 86 | 69,247 | 69,332 | LSC | |
| 12 | (A)11 | 11 | 10,153 | 10,163 | LSC | | 53 | (C)11 | 11 | 70,943 | 70,953 | LSC | |
| 13 | (T)11 | 11 | 10,293 | 10,303 | LSC | | 54 | (T)13 | 13 | 73,588 | 73,600 | LSC | |
| 14 | (T)11 | 11 | 11,217 | 11,227 | LSC | *trnG-GCC* | 55 | (A)10 | 10 | 75,271 | 75,280 | LSC | |
| 15 | (A)11 | 11 | 13,552 | 13,562 | LSC | | 56 | (A)14(A)13 | 34 | 76,829 | 76,862 | LSC | *clpP* |
| 16 | (T)10(A)13(T)11 | 163 | 14,125 | 14,287 | LSC | *atpF* | 57 | (A)11 | 11 | 81,665 | 81,675 | LSC | *PETB* |
| 17 | (T)10 | 10 | 15,319 | 15,328 | LSC | | 58 | (TA)7 | 14 | 83,134 | 83,147 | LSC | *petD* |
| 18 | (A)12 | 12 | 18,667 | 18,678 | LSC | | 59 | (A)10 | 10 | 85,981 | 85,990 | LSC | |
| 19 | (T)10 | 10 | 20,639 | 20,648 | LSC | *rpoC2* | 60 | (T)10 | 10 | 86,299 | 86,308 | LSC | |
| 20 | (T)10 | 10 | 20,768 | 20,777 | LSC | *rpoC2* | 61 | (A)10(T)10 | 36 | 87,375 | 87,410 | LSC | |
| 21 | (T)12 | 12 | 21,296 | 21,307 | LSC | *rpoC2* | 62 | (T)10 | 10 | 89,017 | 89,026 | LSC | |
| 22 | (C)10(A)10(T)10 | 93 | 24,830 | 24,922 | LSC | *rpoC1* | 63 | (T)10 | 10 | 90,643 | 90,652 | IRB | |
| 23 | (T)17 | 17 | 25,303 | 25,319 | LSC | *rpoC1* | 64 | (T)10 | 10 | 114,296 | 114,305 | IRB | |
| 24 | (T)10 | 10 | 28,570 | 28,579 | LSC | *rpoB* | 65 | (T)11 | 11 | 117,274 | 117,284 | SSC | *ndhF* |
| 25 | (T)10 | 10 | 29,642 | 29,651 | LSC | | 66 | (T)15 | 15 | 118,801 | 118,815 | SSC | |
| 26 | (C)13 | 13 | 30,442 | 30,454 | LSC | | 67 | (A)10 | 10 | 118,917 | 118,926 | SSC | |
| 27 | (T)11 | 11 | 31,750 | 31,760 | LSC | | 68 | (A)12t(A)11 | 24 | 119,066 | 119,089 | SSC | |
| 28 | (A)10 | 10 | 32,113 | 32,122 | LSC | | 69 | (T)14 | 14 | 119,222 | 119,235 | SSC | |
| 29 | (A)12 | 12 | 34,229 | 34,240 | LSC | | 70 | (A)12 | 12 | 120,003 | 120,014 | SSC | |
| 30 | (A)13 | 13 | 35,021 | 35,033 | LSC | | 71 | (T)12 | 12 | 122,398 | 122,409 | SSC | |
| 31 | (A)11 | 11 | 36,731 | 36,741 | LSC | | 72 | (A)10 | 10 | 122,745 | 122,754 | SSC | *ndhD* |
| 32 | (A)11 | 11 | 39,921 | 39,931 | LSC | | 73 | (A)11 | 11 | 124,071 | 124,081 | SSC | |
| 33 | (AT)6 | 12 | 40,068 | 40,079 | LSC | | 74 | (T)10 | 10 | 126,004 | 126,013 | SSC | |
| 34 | (T)14 | 14 | 40,210 | 40,223 | LSC | | 75 | (T)11 | 11 | 126,145 | 126,155 | SSC | |
| 35 | (A)13 | 13 | 40,365 | 40,377 | LSC | | 76 | (A)11(T)12(A)11 | 77 | 127,622 | 127,722 | SSC | *ndhA* |
| 36 | (A)10 | 10 | 40,882 | 40,891 | LSC | | 77 | (T)10 | 10 | 130,474 | 130,483 | SSC | |
| 37 | (A)10(A)10 | 89 | 52,317 | 52,405 | LSC | | 78 | (A)12 | 12 | 130,698 | 130,709 | SSC | |
| 38 | (T)11 | 11 | 53,423 | 53,433 | LSC | | 79 | (T)10 | 10 | 133,670 | 133,679 | SSC | *ycf1* |
| 39 | (A)10 | 10 | 53,932 | 53,941 | LSC | | 80 | (T)13 | 13 | 134,247 | 134,259 | SSC | *ycf1* |
| 40 | (T)10 | 10 | 54,316 | 54,325 | LSC | | 81 | (A)10 | 10 | 137,718 | 137,727 | IRA | |
| 41 | (A)10 | 10 | 55,210 | 55,219 | LSC | | 82 | (A)10 | 10 | 161,371 | 161,380 | IRA | |

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
