# Peer review of "Complete Chloroplast Genome Sequence and Phylogenetic Analysis of Quercus bawanglingensis Huang, Li et Xing, a Vulnerable Oak Tree in China"

_forests, doi:10.3390/f10070587_

Round 1

Reviewer 1 Report

The manuscript is good organized and a well written contribution to oak phylogenetics from China with the focus of a newly presented chloroplast genome of an oak species which status is unclear (also taxonomically). I have only few comments (see below) and one recommendation about the conclusion. The conclusion is too ambitious for me. I would appreciate when the authors reduce the conclusion to Asian or even Chinese Quercus species, because that is what they used in this study. Therefore, an overall statement for the genus Quercus is not feasible in this context.

Page 3, line 104, 105: There are a lot of different classifications for the genus Quercus and „section Engleriana“ is a section used only for Chinese oaks. Maybe there is a more adequate citation for the classification of the genus Quercus into sections.

Citation 43: wrong cited – it is Hollingsworth et al. 2009 ( not “Group”) and the citation does not fit into the context of the text where it is cited (page 3, line 105). It is not all about Quercus.

Page 3, line 117, 118: “The cp genome sequences of Q. bawanglingensis have been deposited in GenBank (MK449426).” This sentence belongs to chapter 2.1.

Page 3, line 119-127: This part belongs to an own chapter in Material and Methods called “plant material” not in the introduction.

The text in Figure 3 is not readable.

In Figure 6 it would be helpful when also the sections would be given.

Reviewer 2 Report

The manuscript title “Complete Chloroplast Genome Sequence and Phylogenetic Analysis of Quercus bawanglingensis, a Vulnerable Oak Tree in China” contains information about the oak of the genus Quercus (Fagaceae). The authors performed chloroplast (cp) genome sequencing and assembly. Results revealed that Fagus engleriana and Trigonobalanus doichangensis are closely relative tree, consistent with the fossil records, and that Q. bawanglingensis and Q. tarokoensis, which belong to section Engleriana, share the closest relationship. The analysis of the cp genome of Q. bawanglingensis offers crucial genetic information for further studies of this vulnerable species and of the taxonomy, phylogenetics, and evolution of Quercus. The authors covered the significant information, but they should fix some issues before publication.

Minor issues

Line 115-116: Rewrite the objective as (1), (2), (3) etc...

Line 117-127: this section should be transfer in to the M&M or result.

Line 132: Sampling method need to describe properly, and tree age and variety should add, and at least 1 year history of ecological measures.

Line 139-134: How you quantify the DNA concentration and purity??

Line 134-137: Add the sequencing method briefly.

Major issue

English language need to improve throughout the manuscript.

Author Response

Response to Reviewer 2 Comments

Dear Reviewer,

Thank you for the constructive comments for improving the quality of our manuscript entitled “Complete Chloroplast Genome Sequence and Phylogenetic Analysis of Quercus bawanglingensis, a Vulnerable Oak Tree in China (Forests-532017). We have studied comments carefully and have made correction which we hope meet with approval. The main corrections in the paper and the responds to the comments are as follow.

Point 1:  The analysis of the cp genome of Q. bawanglingensis offers crucial genetic information for further studies of this vulnerable species and of the taxonomy, phylogenetics, and evolution of Quercus. The authors covered the significant information, but they should fix some issues before publication.

Response 1: We think that the analysis of the cp genome of Q. bawanglingensis would offer crucial genetic data for this vulnerable species and the taxonomy, phylogenetics, and evolution of Quercus. The next step is to sample adequately, make further population genetics studies on whether and how natural populations are adapted to their local environment, to predict their response to future habitat alterations and establish adequate conservation strategies for this vulnerable species.

Point 2: Line 115-116: Rewrite the objective as (1), (2), (3) etc...

Response 2: The objective rewrited is as follows (Line 113-119). 

 In this study, we first sequenced and described the complete cp genome of Q. bawanglingensis and performed a comparative analysis of the cp genomes of multiple Quercus species in order to: (1) investigate structural patterns of the whole chloroplast genome of Quercus species including genome structure, gene order, and gene content; (2) examine abundant simple sequence repeats (SSRs) and large repeat sequence in the whole cp genome of Q. bawanglingensis to provide markers for phylogenetic and genetic studies and (3) construct a chloroplast phylogeny for Fagaceae species using their whole cp DNA sequences.

Point 3: Line 117-127: this section should be transfer in to the M&M or result.

Response 3: The content of Page 3, line 117-127 has been moved to Page 3, line 123-132.

Point 4: Line 132: Sampling method need to describe properly, and tree age andvariety should add, and at least 1year history of ecological measures.

Response 4: Sampling method is showed in Line 133-135. The information of this tree is performed in Line 123-132.

Point 5: Line 139-134: How you quantify the DNA concentration and purity?

Response 5: The content about quantifying the DNA concentration and purity has been fed to Page 3, line 135-137, as follows.

 DNA quality was assessed in one drop spectrophotometer (OD-1000, Shanghai Cytoeasy Biotech Co., Ltd., Shanghai, China) and integrity was evaluated using a 0.8% agarose gel.

Point 6: Line 134-137: Add the sequencing method briefly.

Response 6: Sequencing method is:

 ‘Sequencing was performed using an Illumina Hiseq4000 platform (Genepioneer Biotechnologies Co. Ltd., Nanjing, China) with PE250 based on Sequencing by Synthesis (SBS), with at least 5.74 GB of clean data obtained for Q. bawanglingensis.’

Point 7: Major issue: English language need to improve throughout the manuscript.

Response 7:

Thank you for your suggestion. We feel sorry for our poor writing, but we do invite several colleagues who are skilled authors to check the manuscript.

The changes revised are marked in red in text named ‘Forests-532017-track changes.docx’.

And we hope the correction would meet with approval.
